# Estimating Fragmentation and Connectivity Patterns of the Temperate Forest in an Avocado-Dominated Landscape to Propose Conservation Strategies

**María Camila Latorre-Cárdenas** [1,*], **Antonio González-Rodríguez** [1,*], **Oscar Godínez-Gómez** [2], **Eugenio Y. Arima** [3], **Kenneth R. Young** [3], **Audrey Denvir** [3], **Felipe García-Oliva** [1] **and Adrián Ghilardi** [4]

1   Instituto de Investigaciones en Ecosistemas y Sustentabilidad, Universidad Nacional Autónoma de México, Morelia 58190, Mexico
2   Comisión Nacional Para el Conocimiento y Uso de la Biodiversidad (CONABIO), Ciudad de México 14010, Mexico
3   Department of Geography and the Environment, University of Texas at Austin, Austin, TX 78712, USA
4   Centro de Investigaciones en Geografía Ambiental, Universidad Nacional Autónoma de México, Morelia 58190, Mexico
*   Correspondence: mlatorre@iies.unam.mx (M.C.L.-C.); agrodrig@iies.unam.mx (A.G.-R.)

**Abstract:** The rapid expansion of avocado cultivation in Michoacán, Mexico, is one of the drivers of deforestation. We assessed the degree of fragmentation and functional connectivity of the remaining temperate forest within the Avocado Belt and prioritized patches that contribute the most to connectivity using a network-based approach and modelling different seed and pollen dispersal scenarios, including two types of patch attributes (size and degree of conservation). As landscape transformation in the region is rapid and ongoing, we updated the land-use and land-cover maps through a supervised classification of Sentinel-2 imagery, improving the reliability of our analyses. Temperate forest is highly fragmented within the region: most patches are small (<30 ha), have a reduced core-area (28%), and irregular shapes. The degree of connectivity is very low (0.06), dropping to 0.019 when the degree of conservation of patches was considered. The top 100 ranked patches of forest that support the connectivity of seeds and pollen have different characteristics (i.e., size and topology) that may be considered for implementing conservation and management strategies. Seed dispersal seems to be more threatened by fragmentation than pollen dispersal, and patches that are important for maintaining seed connectivity are embedded in the denser zone of avocado orchards.

**Keywords:** structural and functional connectivity; habitat-quality; degree of connectivity; supervised classification; seed and pollen dispersal

## 1. Introduction

Temperate forests constitute the majority of the forest ecosystems in the northern hemisphere and play an important role in supporting ecological processes and nature's contributions to people worldwide [1,2]. In Mexico, temperate forests include pine, oak, pine-oak, and oyamel forests, occupy a large part of the country, nearly 17.4% (340,000 km$^2$; [3]), maintain a high biodiversity of native plants and animals [4–6], and provide fundamental ecosystem services, such as nearly 80–90% of timber production in the country [7] and about 54% of the carbon sequestration and 25% of the water infiltration [8–10].

Mexico is one of the centers of the diversification of pines (*Pinus*, Pinaceae; [11]) and oaks (*Quercus*, Fagaceae; [12]), harboring 46 of 110 pine taxa and 160 of about 450 oak species that have been described worldwide [4,13,14]. Despite the invaluable biological heritage and the ecological importance of temperate forests for Mexico, these forests have been lost at an annual rate of 0.5–0.8%, mainly due to changes in land use, increased fires, and illegal logging [5,6,13]. In the state of Michoacán, deforestation has led to the loss of up

to eight thousand hectares of temperate forests per year, reaching an annual deforestation rate of 1.8% in the period of 1975–1993 [15], which decreased on a statewide scale (from 0.16 to 0.09% between 2007 and 2014) but increased at specific municipalities located at the center of the state [16–18]. These deforestation dynamics have been driven mainly by the opening of avocado exports in Michoacán to the United States of America, which multiplied sixty times from 2000 to 2018, producing 60% of the fruit purchased in the USA and leading to a strong impact on biodiversity, such as the impoverishment of amphibian communities [19], changes in pollination rates and in the distribution of pollinators [20,21], the degradation of soil, and a decrease in water retention [22] and alteration of hydrological systems [23,24]. On the other side of the coin, the avocado industry has led to some economic benefits (i.e., generation of employment and a reduction of out-migration), although social inequality in the region still limits these benefits [24].

It has been estimated that between 2001 and 2017, 20% of the deforested areas in Michoacán were replaced by avocado cultivation [23]. Currently, there are no updated deforestation rates for the state, but the reduction in forest cover by avocado cultivation continues to be reported in some municipalities [21,25], and the area planted with avocado increased at an annual rate of 4.5% from 2013 to 2021 [26].

Recently, Arima et al. [27] modeled the distribution of the likely future expansion of avocado plantations in Michoacán by 2050 and reported a high deforestation risk of nearly 1000 km$^2$ across pine-oak, mesophilic montane, and oyamel fir forests. This is a matter of concern because the constant loss of habitat areas promotes the fragmentation of continuous forest into small and isolated patches, resulting in lower diversity and decreased dispersal of seeds and pollen among patches, thereby impacting the natural connectivity patterns of plant populations [28].

Connectivity is defined as the degree to which a landscape facilitates or impedes the movement of individuals, propagules, and ecological flows [29,30]. The loss of habitat and connectivity during fragmentation can reduce the carrying capacity of patches, leading to lower establishment and survival probabilities of seedlings and saplings [31,32]. Furthermore, fragmentation and the loss of connectivity not only affects the composition, distribution, and persistence of the forest itself but also the ecological processes and the diversity of the species that depend on it [33,34]. Connectivity can be assessed by considering the spatial relationship among landscape structural elements (structural connectivity), as well as the response that organisms have to the arrangement of those elements (functional connectivity), which largely depends on their dispersal capacities [35]. Graph-based connectivity metrics are useful for assessing the connectivity of habitat patches in a landscape and for spatial prioritization [36]. A graph or network consists of nodes (habitat patches characterized by attributes such as size and quality) that are connected by links and surrounded by a matrix of non-habitats [37]. Graph-based metrics, such as the integral connectivity index (IIC; [38]) are more robust than other connectivity metrics because they consider the area of the habitat within patches (intra-patch connectivity) and the area that can be reached by links between patches (inter-patch connectivity [36,38]. Regarding fragmentation metrics, they facilitate the quantification of changes in the spatial structure of habitat patches and the understanding of the relationships between different patches present in the landscape [39].

Having a measure of the effective dispersal of plants is a challenge because it involves the production of both seeds and pollen at a source patch, their movement by biotic and abiotic vectors to other habitat patches, and the successful establishment of new individuals at the recipient patch [40]. These processes are affected by the dispersal capacity of seeds and pollen, heterogeneity of the landscape, and attributes of the patches associated with the survival of seeds, such as their size, habitat quality-weighted area, habitat suitability, and core area [41]. For example, larger patches with a better quality have a higher carrying capacity and more available resources and dispersal agents [42]. Dispersal distances of pine and oak species differ between seeds and pollen. Seeds are dispersed by small mammals (i.e., squirrels and mice) and some birds, reaching distances of up to 1500 m, but usually

not exceeding 500 m [43,44]. Pollen, which is dispersed by wind [14,45] frequently reaches distances greater than 1 km [46,47], but distances of 30 km and up to 80 km have been reported [45,48].

In that context, we assessed the degree of fragmentation and functional connectivity of the remaining temperate (mainly pine-oak) forests within the region of avocado cultivation in Michoacán. We prioritized key patches that contribute the most to connectivity, which is essential information needed to propose forest conservation strategies that help to mitigate the negative effects of fragmentation and habitat loss. We also evaluated the effect of the different dispersal capacities of seeds and pollen on connectivity patterns and the importance of considering habitat quality as a patch attribute. To accomplish this, we employed a network-based approach and modeled different scenarios for which the dispersal distance capacity of the pine and oak species (1, 5, 10 and 20 km) and the attributes of patches (size and the degree of conservation) varied. Our hypotheses were that we would find different patterns of connectivity mediated by seeds or by pollen, and a decrease of connectivity when habitat quality weighted by area is considered as a patch attribute. As landscape transformation in the region has been rapid, first we updated the land use and land cover (LULC) maps through a supervised classification method, using a set of Sentinel-2 imagery. Once we evaluated the connectivity of the forest under different scenarios, we identified the patches that contribute the most to connectivity, based on a novel connectivity composite index and the importance of the patches to maintain flux and connection within the avocado-affected landscape.

## 2. Materials and Methods

The study area corresponds to the avocado growing region of the state of Michoacán (hereafter Avocado Belt; Figure 1), which according to official data includes 167,748 hectares of avocado orchards and covers 46 municipalities in the state [26]. To this area, we added a 15 km buffer to include other forest patches that possibly form part of the functional connectivity. The total area of the analyzed landscape was 3,995,800 ha. The study area has great topographical and climatic heterogeneity, with an altitudinal gradient between 1300 and 3600 m, different geomorphological formations (i.e., mountains, plateaus, valleys, and hills), and semi-warm sub-humid and temperate sub-humid climates, with a marked rainy season from June to September, with annual means for temperature and precipitation ranging from 10 to 24 °C, and 800 to 1600 mm, respectively [49]. The Avocado Belt has a variety of soil types, with andosols, luvisols, and acrisols being the most representative and adequate for agriculture, especially for the cultivation of avocados [22,27,50]. Temperate and tropical forests are the types of vegetation that dominate the region, followed by agricultural areas consisting mainly of avocado, berries, maize, sorghum, wheat, guava, agave, and peach [7]. The human population within the Avocado Belt is approximately 4,036,400, the largest cities are Morelia (1,364,000 ha) and Uruapan (452,800 ha), and the region is well connected by major highways [7].

### 2.1. Land Use and Land Cover Classification and Accuracy

LULC classification was performed using the random forest (RF) classifier, a supervised method that is based on the iterative and random creation of multiple decision trees [51]. RF uses randomly selected subsets of training samples and variables that correspond to spectral bands of satellite images and even other ancillary data such as elevation [51]. In this analysis, we employed ten Sentinel-2 scenes [52] to cover the footprint of our study area, taken between March and May 2019 because, in those months, the drought made it easier to distinguish between the forest trees that were somewhat dry compared to the artificially hydrated avocado crops. Sentinel-2 images have resolutions of 10 and 20 m and consist of ten spectral bands (B2-B3, B4, B5, B6, B7, B8, B8A, B11, B12). For each scene, atmospheric and radiometric rectifications were performed using the Dark Object Subtraction (DOS1) method and the reflectance metadata file of images, both using the preprocessing correction tools of the Semi-automatic Classification Plugin [53] for QGIS

software [54]. We classified each scene separately and then we merged all scenes into a single mosaic. RF classification was performed using the Semi-automatic Classification Plugin [53]. We created several training areas, called regions of interest (ROI), which are polygons drawn over homogeneous areas of the image that overlay pixels belonging to the same land cover class. Each ROI was assigned to a land cover class, using as references Google Earth images for 2019 and the land use and land cover layer created in 2014 for Michoacán [55]. We created 15 to 20 ROIs for each of the following classes: (1) *temperate forest* (TF) which included pine, oak, and pine-oak forests, and a small percentage corresponding to oyamel fir forests, (2) *avocado orchards* (AVO), (3) *water bodies* (WB), mainly lakes and reservoirs, (4) *agriculture and pasture areas* (AP), which included all types of croplands, except the avocado orchards, (5) *settlements* (SET), that consisted of urban zones and roads, and (6) *dry forest and shrublands* of low and medium deciduous forests (DF). We calculated the spectral signature for each ROI and used them to perform the RF classification, using 5000 points (pixels) for training the model and running 120 decision trees, as suggested by Congedo [53].

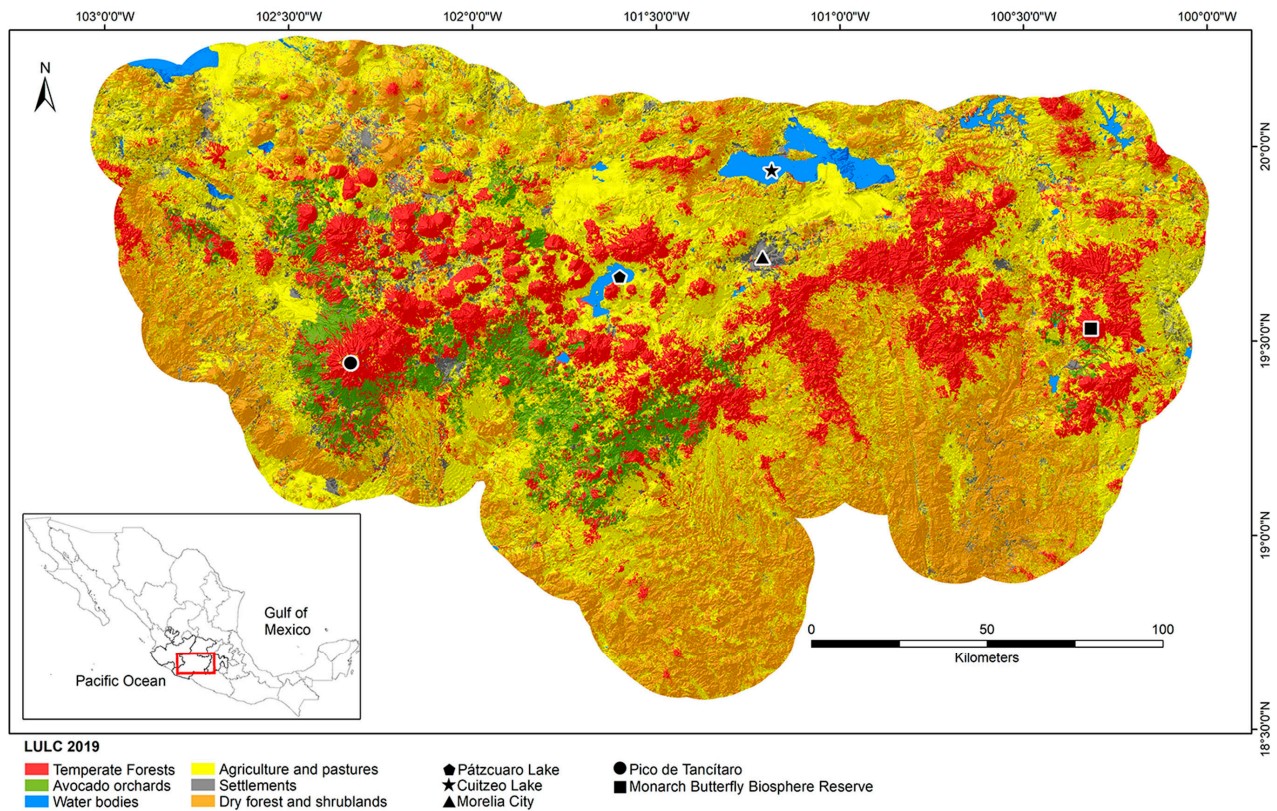

**Figure 1.** Land-use and land-cover map (2019) of the Avocado Belt, located in the state of Michoacán, Mexico. The inset on the bottom left shows the geographic location of the study area (red box) within the country.

Once we obtained the mosaic of the classified Sentinel scenes, we used the postprocessing tools of the semi-automatic Classification plugin: (1) "edit raster" to correct pixels that were misclassified, especially some pixels of the forest class on slopes that were misclassified as avocado, or some alfalfa and wheat crops that were misclassified as avocado and (2) "classification sieve" to filter and remove speckling, replacing the class values of isolated pixels with the class value of the largest neighboring patch.

For the accuracy assessment of the classification map, we performed stratified random sampling, following Olofsson et al. [56] and Olofsson et al. [57]. Briefly, we created a sample of reference sites (e.g., pixels), using the map categories as strata. Then, we constructed a confusion matrix (also known as the error matrix), with the map categories and reference

categories represented by rows and columns, respectively. When using stratified sampling, the number of samples for each mapping category is not necessarily proportional to the area covered by each category. Therefore, we adjusted the confusion matrix by weighting the number of sites using the area of each category on the map, as suggested by Card [58]. Using the values of the adjusted matrix, we calculated (1) the overall accuracy, which is the overall proportion of area correctly classified, (2) the user accuracy, which is related to omission errors and indicates the proportion of the reference sample, for a particular category, that is correctly classified in the map, and (3) the producer accuracy, that is related to the commission error, and shows the proportion of samples classified as a particular category in the map which are correctly classified [55,59]. Confidence intervals (CI) for each accuracy measure were estimated using the bootstrap percentile interval method. Analyses were performed in Dinamica Ego [60].

### 2.2. Fragmentation and Connectivity of the Temperate Forest

To assess the degree of fragmentation and connectivity of the temperate forest within the Avocado Belt, we used the *temperate forest* class data of the LULC map and selected the patches that had an area equal to or greater than 10 ha for downstream analysis. This criterion was used to reduce the number of patches classified as temperate forest (initially $n = 23,225$) and handle an adequate number of nodes to build the connectivity graphs, and to avoid including in the prioritization analysis patches that, being too small, do not have a core area. Fragmentation metrics that we calculated were the size of the forest patches, area, and percentage of the core and edge of patches, as well as the shape index and perimeter-area ratio (PARA). For estimating the core and edge area, we considered 50 m as the 'edge distance' based on the reports of the regeneration of pine and oak species within fragmented temperate landscapes [61,62]. Shape index (SI) indicates how irregular a patch is, considering a patch to have a regular shape when it is circular (value 0) and increasing ($\geq 1$) when the patch shape becomes more irregular [63]. PARA relates the area and perimeter of a patch, increasing when the complexity of the patch shape is higher [63]. We estimated fragmentation metrics using the function *MK_Fragmentation* of the *Makurhini* package [64] in the R environment and programming language v.4.1.3. [65].

We assessed connectivity using two scenarios based on the dispersal capacity of pine and oak species that are supported by genetic studies and restoration monitoring. The *seed dispersal scenario* considered a distance threshold of 1 km, which is the maximum that can be reached by seeds [42–44]. The *pollen dispersal scenario* included distances of 5, 10, and 20 km that can be reached in some cases by pollen [46,47]. To quantify temperate forest connectivity patterns and prioritize habitat patches, we estimated the integral index of connectivity, which is based on binary connections between patches (presence or absence) and considers the attribute of patches, allowing us to incorporate into the model the intrapatch connectivity (IIC; [38]). We chose IIC among other connectivity metrics, such as the probability of connectivity index (PC; [66]), because we used specific values as the maximum dispersal distance that seeds and pollen can reach. Moreover, IIC has shown a stronger relationship with empirical data on ecological patterns and processes [36,67]. We estimated the IIC for the entire landscape and for each habitat patch, hereafter *focal habitat patch* ($f$). For each $f$ we iterated the selection of neighboring habitat patches using a search radius (or buffer) that was twice the scenario of dispersal distance thresholds (i.e., 2 km of buffer when dispersal distance was 1 km). Habitat patches within the search radius were selected and classified as *transboundary habitat patches* (*thp*), which might have the highest probability of connectivity with the focal habitat patch. Then, for each focal habitat patch selection and its transboundary patches in each iteration, we calculated the IIC as follows:

$$IIC_f = \frac{\sum_{i=1}^{f+thp} \sum_{j=1}^{f+thp} \frac{a_i a_j}{1+nl_{ij}}}{A_L^2} \qquad (1)$$

where $a_i$ and $a_j$ are the attribute value of the focal habitat patch $f$ and the transboundary patches *thp*. In this study, we estimated the IIC$_f$ using two attributes of the patches: the habitat area and a proxy of the degree of habitat conservation. The latter refers to the areas of patches $f$ and *thp* weighted by their degree of conservation, which was obtained from the index of human impact on the terrestrial biodiversity of Mexico (MEXBIO; [68]). The MEXBIO index ranges from 0 to 1, where 1 indicates high human impact. It is based on the theoretical framework of the Global Biodiversity Model (GLOBIO 3; [69]) and includes pressure factors and threats from land use, road infrastructure, and fragmentation data [68]. We inverted the range of the index (MEXBIO value—1), extracted the average value within each patch, and then multiplied it by the area of the patch to have a measure that represents how conserved (less transformed) the habitat patches are. By using both patch attributes in the IIC models, we assumed that the larger the patch area and the more conserved, the greater the intra-patch connectivity and the contribution to connectivity. $nl_{ij}$ is the number of links within the shortest path from patch $i$ to $j$ and was estimated considering the Euclidean distance between habitat patches and the four dispersal distances thresholds established for the *seed* and *pollen scenarios*. *AL* is the landscape area and was estimated by considering the area of the landscape for each focal patch, generated in each iteration (i.e., the extent of $f$ and *thp*).

We prioritized and ranked focal habitat patches ($f$) by calculating their contribution to the network connectivity (dIIC$_f$), which corresponds to the percentage of the variation in IIC$_f$ caused by the removal of each individual patch ($f$) and *thb* from the landscape of each focal patch [70,71]. dIIC$_f$ can be divided into three fractions considering the different ways in which a certain landscape element (patch or link) can contribute to habitat connectivity and availability in the landscape depending on the attribute of the patch, the topological position of the patch within the landscape, and the dispersal capability of the focal species:

$$dIIC_f = dIIC_{f\ intra} + dPC_{f\ flux} + dPC_{f\ connector} \tag{2}$$

where dIIC$_{f\ intra}$ is the contribution of patch $f$ to the intrapatch connectivity, in other words, the contribution to the availability of habitat based on its attribute. It does not depend on the dispersal capacity of the focal species, the topological position, or the intensity of connections. Instead, dIIC$_{f\ flux}$ corresponds to the patch attribute-weighted dispersal flux through the connections of patch $f$ to or from all of the other patches in the landscape when $f$ is either the starting or ending patch of that connection or flux [38,66]. dIIC$_{f\ flux}$ depends both on the attribute of patch $f$ and its position within the local landscape network [38,66]. dIIC$_{f\ connector}$ is the contribution of patch $f$ to the connectivity as a connector element (also known as a stepping-stone), only considering its topological position within the landscape of the focal patch [38,66].

We estimated a Composite Connectivity Index (CCI$_f$) as a tool to prioritize each focal patch based on its individual contribution to connectivity in the $f$ and *thp* patch network (dIIC$_f$) and the landscape connectivity of the entire network (IIC$f$). In that sense, patches with higher CCI values are in a well-connected landscape and their contribution to connectivity is considered important:

$$CCI_f = IIC_f * dIIC_f \tag{3}$$

To prioritize the most important patches for maintaining forest connectivity within the Avocado Belt, we selected the 100 best ranked patches for CCI$_f$, dIIC$_{f\ flux}$, and dIIC$_{f\ connector}$.

We calculated the Equivalent Connectivity Area (ECA) for each *focal habitat patch*, which is defined as the size of a single habitat patch (maximally connected) that would provide the same value of connectivity as the actual habitat pattern in the landscape [72]. ECA$_f$ was derived from the IIC and defined as:

$$ECA_f = \sqrt{\sum_{i=1}^{f+thp} \sum_{j=1}^{f+thp} \frac{a_i a_j}{1 + nl_{ij}}} \tag{4}$$

where $a_i$ and $a_j$ are the attribute values of the focal habitat patch $f$ and the transboundary patches *thp*; $nl_{ij}$ is the number of links within the shortest path from patch $i$ to $j$ and was estimated considering the Euclidean distance between habitat patches and the four dispersal distances thresholds established for the *seed* and *pollen scenarios*.

Finally, we performed a mixed-effect model to assess whether the connectivity index and the contribution of forest patches (response variables) differed between the dispersal capacity distances of seeds and pollen, and the attribute of the patches (fixed effects). We used the identity of the patches as the random effect. This analysis was performed using the *nlme* function in R.

## 3. Results

### 3.1. Land Use and Land Cover Classification and Accuracy

Through the Random Forest classifier, we were able to obtain a 2019-LULC map of the Avocado Belt in the Michoacán state, which consists of six classes and has a 30 m resolution (Figure 1). Temperate forests have an extension of 786,812 ha and avocado orchards 244,705 ha, being around 20 and 6% of the entire Belt area, respectively (Table 1). Forests are located along the entire Belt, and avocado orchards are mainly in the west and central zones, and in a small extension to the east. Agriculture and pasture form the class with the largest extension, followed by dry forest and shrublands (Table 1). Settlements and water bodies occupied the least area (Table 1).

**Table 1.** Classes of the land-use and land-cover map for the Avocado Belt in the state of Michoacán, Mexico. The extension and percentage of the landscape that represents each of the classes are presented. User (UA) and producer accuracy (PA), with the corresponded confidence intervals (CI) are shown. Overall accuracy of the classification is included.

| Class | Area (ha) | % Land | UA (%) | CI (%) | PA (%) | CI (%) |
|---|---|---|---|---|---|---|
| Temperate forests | 786,812 | 19.8 | 91.11 | 4.16 | 0.59 | 0.04 |
| Avocado orchards | 244,705 | 6.1 | 65.61 | 7.43 | 0.64 | 0.11 |
| Water bodies | 79,621 | 2.0 | 91.45 | 5.07 | 0.85 | 0.13 |
| Agriculture and pastures | 1,590,748 | 40.0 | 55.63 | 5.78 | 0.78 | 0.04 |
| Settlements | 180,329 | 4.5 | 31.29 | 7.50 | 0.77 | 0.20 |
| Dry forest and shrublands | 1,097,680 | 27.6 | 75.96 | 5.81 | 0.68 | 0.05 |
| Overall | 3,979,896 | 100 | 68.40 | 2.98 | - | - |

A total of 1093 verification points were used to assess the accuracy of the model, with a different number of points per class or strata ($n$ = 292-TF, 125-AVO, 113-WB, 273-AP, 294-SET, and 246-DF). The overall accuracy of the classification was 0.685 (CI 0.655–0.715) and the user accuracy ranged from 0.31 to 0.91, with settlements being the class with the lowest value and temperate forest the class with the highest (Table 1). Producer accuracy values were lower than user accuracy values (Table 1).

### 3.2. Fragmentation and Connectivity of the Temperate Forest

The assessed landscape consisted of 2433 patches of temperate forest, as classified by Random Forest, ranging from 10 to 182,562 ha in size, with a mean of 301 ha (Figure S1a). Most of the patches (90%) had an area smaller than 30 ha. Large patches of forest are mainly located in the east, while smaller patches occur at the west and center of the region, surrounded by patches of avocado orchards (Figure 1). The percentage of the core and edge areas that represented all patches of the landscape was 74.9 and 25.1%, respectively. Most of the forest patches had a reduced core area (mean 28%) due to large edge areas. For example, only 38% of patches had at least 20% of core area and only 8.7% had more than 50% of core area (Figure S1b,c). The shape index varied from 0.37 to 288,259, with a mean of 247.8. Only 16.3% of the patches had a Shape I value < 1 and 50% a value < 2 (Figure S1d). PARA varied from 2.09 to 28.82, with a mean of 5.27 (Figure S1e).

The connectivity index $IIC_f$, the composite connectivity index ($CCI_f$), and the Equivalent Connectivity Area ($ECA_f$) differed depending on dispersal distance thresholds ($F = 1856.8$, $p \leq 0.001$, $F = 141.8$, $p \leq 0.001$, $F = 15,989$, $p \leq 0.001$, respectively) and patch attributes ($F = 1045.5$, $p \leq 0.001$, $F = 92.7$, $p \leq 0.001$, $F = 7325.3$, $p \leq 0.001$, respectively, Table S1). The contribution of the patches to connectivity $dIIC_f$ and its fractions ($dIIC_{f\,intra}$, $dIIC_{f\,flux}$, $dIIC_{f\,connector}$) differed with dispersal distance ($F = 1255.4$, $p \leq 0.001$), but not in relation to the size and degree of the conservation of patches ($F = 0.31$, $p \leq 0.578$). The general pattern of change for these indices was the reduction in connectivity and the contribution of values of local patches when the dispersal distance increased, or the reduction in connectivity of the landscape when the degree of conservation was considered as the attribute of the patches (Table S1).

The integral index of connectivity (IIC) for the entire region was 0.087, 0.095, 0.164 and 0.242 for 1, 5, 10 and 20 km dispersal distances, respectively. The mean of the integral index of connectivity calculated for local patches ($IIC_f$) was low (0.06) for the *seed dispersal scenario*, dropping to 0.019 as dispersal distance increased for *pollen dispersal scenarios* (Table S2, Figure S2a). When the degree of conservation of the patches was considered as the patch attribute, $IIC_f$ was 0.02 for the *seed dispersal scenario*, and 0.003 for the 20 km *pollen dispersal scenario* (Table S2). Most of the patches of forest contributed to connectivity at a low percentage (mean of $dIIC_f = 8.04\%$ when patch size was considered, and 7.96% when the degree of conservation was considered). However, some patches reached an importance of 70 to 99% in their local network (Table S2, Figure S2b). These important patches were mainly of medium size (150–1500 ha) and were located in the western and eastern parts of the region in the *seed dispersal scenario* (Figure S2b), while for the *pollen dispersal scenarios*, the important patches were of larger size (>1500 ha) and located in the east of the Avocado Belt (Figure S2b). The fraction $dIIC_{f\,intra}$ had similar values to $dIIC_f$, in both seed and pollen dispersal scenarios (Table S2; Figure 2a). $dIIC_{f\,flux}$ and $dIIC_{f\,connector}$ indicated that the smallest patches, located in the peripheries of the region, are the ones that promoted flux and act as stepping-stone elements in the *seed dispersal scenario* (Figure S2b,c). In contrast, when the maximum potential dispersal of pollen was considered (20 km), the majority of patches act as sources or receivers of flux, that is, a high flow is maintained in the landscape, and the patches that act as stepping-stones are located in the center of the region (Figure 2b,c).

The prioritization of the patches of forest, using the $CCI_f$, indicated that patches located in the northwest and east of the Avocado Belt are the most important in the *seed dispersal scenario* (Figure 3a). For the *pollen dispersal scenarios*, in addition to the aforementioned patches, those located in the center formed a belt of well-connected patches throughout the Avocado Belt (Figure 3a). For the *seed dispersal scenario*, the top 100 ranked patches for $CCI_f$ are distributed around the whole Avocado Belt (Figure 4a), except in the central zone. The top-ranked patches for $dIIC_{f\,flux}$ and $dIIC_{f\,connector}$ are well distributed across the region, but they are very small patches (Figure 4b,c). For the *pollen dispersal scenario*, the top 100 ranked patches for $CCI_f$, are evenly distributed across the Avocado Belt and they are almost the same for the 5 and 20 km scenarios (Figure 4a). The top-ranked patches for $dIIC_{f\,flux}$ and $dIIC_{f\,connector}$ are mostly located in the northwest zone for the 5 km scenario. For the 20 km scenario, patches that maintain flux are distributed across the region and those acting like stepping-stones are found in the center (Figure 4b,c).

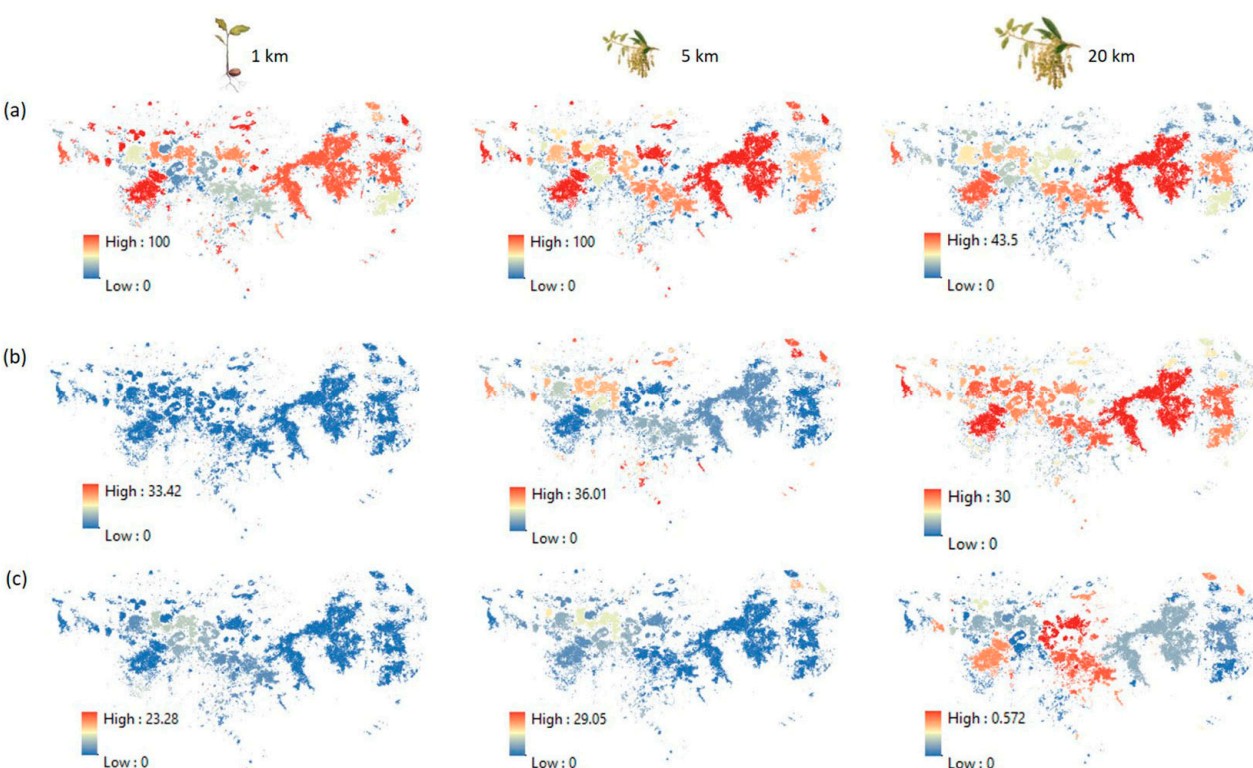

**Figure 2.** Maps of the temperate forest within the Avocado Belt of Michoacán, Mexico, showing the dIIC$_{f\,intra}$ (**a**), dIIC$_{f\,flux}$ (**b**) and dIIC$_{f\,connector}$ (**c**). Indices are shown for the dispersal distance thresholds of seeds (1 km) and pollen (5 and 20 km), considering the size as the patch attribute. Values of the indices are indicated by the color of the bars.

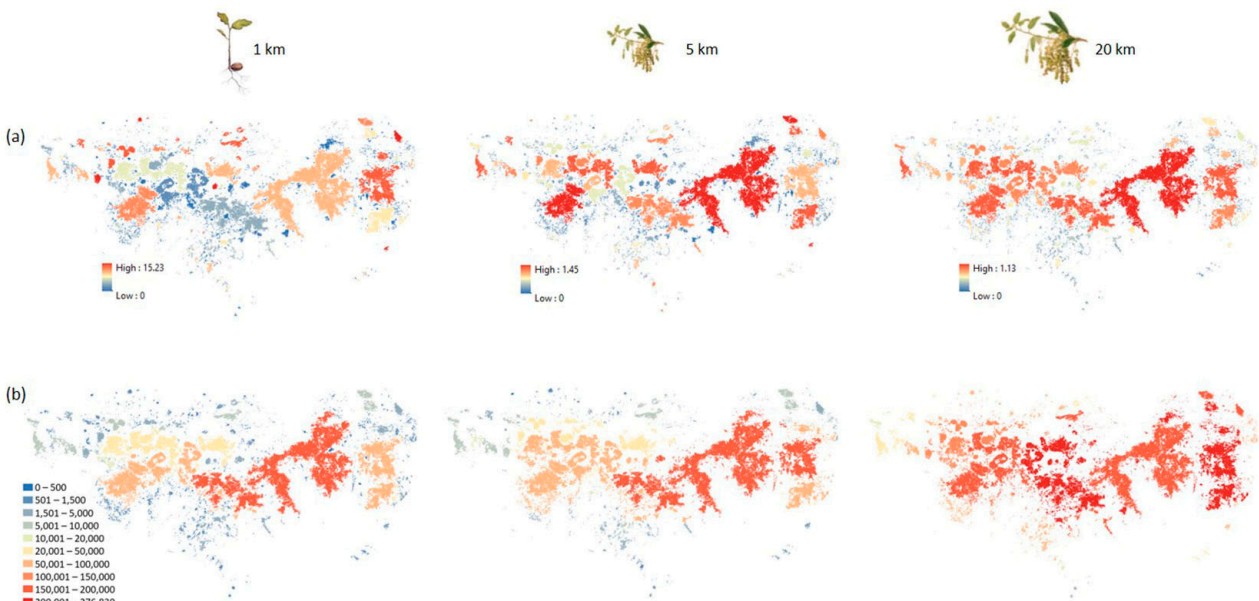

**Figure 3.** Maps of the temperate forest within the Avocado Belt of Michoacán, Mexico, showing the prioritizing indexes: CCI$_f$ (**a**) and ECA$_f$ (**b**). Indices are shown for the dispersal distance thresholds of seeds (1 km) and pollen (5 and 20 km), considering the size as the patch attribute. Values of the indices are indicated by the color of the bars.

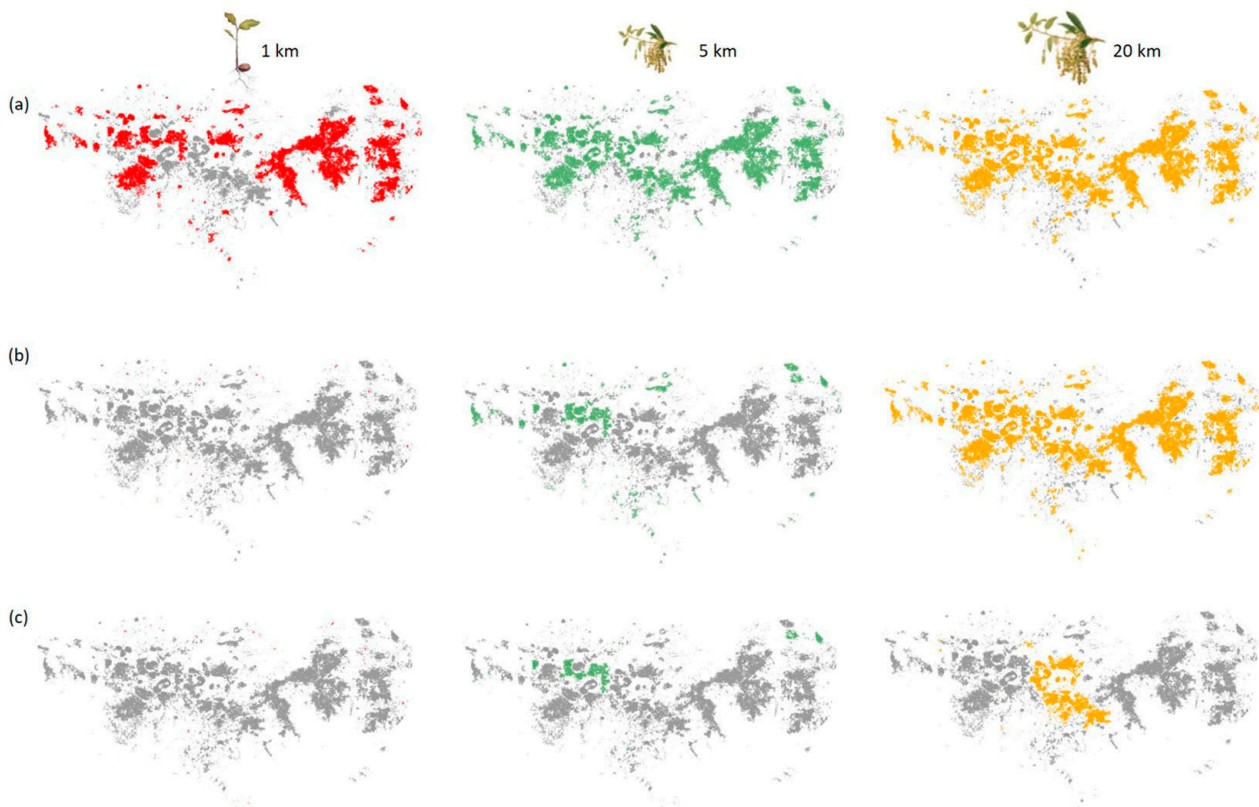

**Figure 4.** Maps of the top 100 ranked patches of temperate forest within the Avocado Belt, based on the CCI$_f$ (**a**), dIIC$_{f\,flux}$ (**b**) and dIIC$_{f\,connector}$ (**c**). Top patches are shown in red for the seed dispersal scenario (of 1 km), and in green and orange for the pollen dispersal scenarios, respectively. All the patches of temperate forest that occur within the Avocado Belt are in gray.

Equivalent Connectivity Area (ECA$_f$) reached a maximum value of 276,830 ha of potentially connected areas of forests in the 20 km *pollen dispersal scenario* (Table S2; Figure 3b). This means that under this scenario, up to 37.74% of the forest can remain connected. The maximum value of the ECA$_f$ index in the 5 km *pollen dispersal scenario* was 199,079 ha (27.14%), and 194,126 ha (26.46%) for the *seed dispersal scenario* (Table S2; Figure 3b). In general, patches located along the central part of the region had high values of ECA$_f$ (Figure 3b). ECA$_f$ values were lower when the degree of conservation of forest patches was considered, reaching a maximum value of 122,205 ha in the 20 km *pollen dispersal scenario*, which represents 15.30% of the area weighted by the conservation degree, 90,217 ha (12.30%) at the 5 km *pollen dispersal scenario*, and 88,112 ha (12.01%) for the *seed dispersal scenario* (Table S2).

## 4. Discussion

The 2019-LULC map that we developed helped us to accurately assess the expansion of agriculture, mainly avocado cultivation, and the degree of fragmentation and connectivity of the temperate forest within the Avocado Belt of Michoacán. It shows the spatial distribution of six land cover classes with a high resolution and a good overall accuracy of 0.69. The temperate forest class had high user and producer accuracies, supporting confidence in our fragmentation and connectivity analyses. Likewise, the avocado orchards class had a moderate to high accuracy, allowing us to quantify and update the extent of these crops within the avocado-growing region. The lower values of users' accuracy for settlements and agriculture-pasture classes indicate a confusion of these classes with other land cover classes. This may be because there are isolated trees or lines of trees within the urban areas that can confuse the classification algorithm, as well as greenhouses that cannot be distinguished from buildings.

We calculated from the 2019-LULC map a total of 244,705 ha of avocado cultivation, which is considerably higher than the reported extension for 2019 (167,747 ha) and 2021 (174,442 ha), according to Mexico's Agrifood and Fisheries Information Service (SIAP in Spanish [26]). This difference suggests that more than 75,000 ha of avocado cultivation are not officially registered with the respective government entities, which may be because some orchards are not itemized until they are productive, in a period of 3 to 5 years, or because they are established after illegal logging of the forest. Under any of the plausible scenarios, knowing the extension of avocados is critical to have adequate management and estimate its potential ecological and social consequences. For example, the current area of avocado cultivation must represent a higher annual rate extension than the reported (4.5%; [26]), with consequences for the water footprint of this crop. It is well documented that avocado production, under dry conditions, can consume up to 120% of the surface and groundwater volumes granted to other crops, leading to water stress and social conflicts [73].

Loss and fragmentation of the temperate forest in Michoacán have been caused by the cultivation of avocados at a significant percentage (20%; [23]). For example, in the Meseta Purépecha, which is located in the central-west zone of the avocado region, saw an increase of 10.8% in agriculture coverage, mainly of avocado orchards, and a forest reduction of 15% was identified during the period 1986–2016 [18]. Nonetheless, other agriculture types, which have an important extension in the region (40%), may also be contributing to the conversion and connectivity loss of the forests. For example, berry production in Michoacán has increased given its high profitability at the national and international levels [74]. However, there are still no studies that evaluate their impacts on forest connectivity.

Metrics of fragmentation indicate that the temperate forest within the Avocado Belt is now highly fragmented as the majority of patches are smaller than 10 ha and several of those that are larger have a reduced core area (50%), meaning that they potentially contribute less to connectivity than would be expected if the total area of the patch was available as an interior habitat. Shape-metrics show that patches are irregular, with large edge areas, which is further evidence that fragmentation is high. The edge effect is an important factor to consider when estimating the connectivity and regeneration patterns of the forest in the region. For example, the seedling abundance of oaks can differ between the edge and interior of fragments, depending on the light, temperature, and humidity requirements of species [62], and the configuration of edge ecotones (i.e., transition interior-edge-matrix) also influences the regeneration process of oaks [43].

The degrees of connectivity (IIC) among all the forest patches within the Avocado Belt were low, being lower (0.087) for the *seed dispersal scenario* and reaching a maximum value (0.242) in the 20 km *pollen dispersal scenario*. In contrast, the degree of connectivity for the local networks of patches (IIC$_f$) was low and decreased as dispersal distance increased. This means that as the size of the search ratio increases (because of a larger dispersal distance), the number of *transboundary neighboring patches* increases and, in consequence, *local patches* are more isolated, reducing the overall connectivity value (IIC$_f$). This makes sense considering that the study area's forest is highly fragmented, mainly composed of small patches. It is also important to emphasize that the absolute values of IIC are dependent on the definition of the boundaries of the study area (area of landscape), resulting sometimes in anomalous low values when the habitat patches and total habitat area are small compared to the entire landscape [66]. However, we avoided this by considering that not all the patches that compose the landscape are interconnected, but rather are small networks of *focal patches* that are potentially connected depending on the dispersal capacity of a species. Therefore, it is also relevant to apply these connectivity metrics using different dispersal scenarios since patches that are not important for organisms with high vagilities may be for organisms with a restricted vagility. The degree of connectivity in the study area is similar to the probability of connectivity (PC = 0.023) reported for a smaller network of 510 patches of oak forest, located in the northern part of the avocado region, around the

Cuitzeo lake [75]. In the same area, low connectivity (IIC) has also been reported for two medium-size species of mammals with moderate dispersal capacities (10 km; [76]).

The dIIC$_f$ metrics showed the different contributions to connectivity that each forest patch has, which depended on the dispersal distance capacity. Patches that contributed the most to the availability of habitat (higher values of dIIC$_{f\ intra}$) were of medium and large size, but their importance decreased as the dispersal distance increased. This is because when a species has low mobility, the intrapatch connectivity (availability of area in the patches) is more important for its survival and movement than when it is highly dispersed and can reach other available patches in the landscape [66,77]. On the other hand, patches that were repeatedly at the start or end of a short path (high dIIC$_{f\ flux}$) or that acted as stepping-stones within the path (high dIIC$_{f\ connector}$) were of small and medium size and located at the peripheries of the region in the *seed dispersal scenario*, while for the *pollen dispersal scenarios*, patches that contributed to connectivity were of a large size and located along a centered transverse strip. These patterns are expected because when mobility is restricted, patches that receive more flux are those near other habitat patches and not necessarily the largest ones [66]. As we mentioned before, the study area's forest is highly fragmented, thus it is likely that patches close to each other are small. In contrast, with an increased ability to disperse large distances, as pollen does, relevant patches coincide with those that are the largest within the landscape [66,78].

Patches that contributed the most to the connectivity in the *seed dispersal scenario*, either due to the habitat they provide or their position within the flow networks, were found surrounding the central-western zone where most of the avocado orchards occur and at the east where the largest and most continuous patch of forest is situated. This suggests that if the loss and fragmentation by the avocado cultivation continues, the disruption of seed dispersal would have negative effects on the recolonization and regeneration of the central-western forest. The continuum of forest at the east is an important element of the landscape that is potentially threatened by the predicted expansion of avocados towards the east of the Avocado Belt by 2050 [27]. Therefore, its conservation is critical for supporting the long-term connectivity of the temperate forest. Otherwise, it seems that conserving the large and centrally located patches can ensure continued pollen dispersal. In the region, high levels of gene flow and low genetic differentiation are reported for oak and pine species [79–81]. For one oak species (*Quercus castanea*), it was documented that gene flow rates mediated by pollen are much higher than those of seeds [82]. This can be attributed to the higher connectivity among large patches that facilitate pollen dispersal.

We implemented the Composite Connectivity Index (CCI$_f$) to prioritize patches, considering that they should be in a well-connected landscape and have a high contribution to connectivity. If we take the 100 most important forest patches for both the seed and pollen dispersal scenarios, it gives us an evenly dispersed forest cover within the Avocado Belt, where the northwestern and eastern patches are important for seed dispersal and the same ones plus the central patches for pollen dispersal. Therefore, we propose that this set of patches can be considered as critical components of forest conservation. Moreover, this set of patches is part of a biological corridor that connects two nationally protected areas (Pico de Tancítaro and the Monarch Butterfly Biosphere Reserve) and is considered as a priority for conservation [83].

We found that large patches of forest are the most relevant for maintaining a large amount of connected habitat (high values of ECA$_f$), and that in general, patches located along the middle zone of the study area had the higher values. When comparing the pollen and seed scenarios, we detected a decrease of maximum connected forest area from 276,830 to 194,126 ha, suggesting that, in the seed scenario, about 10% of the connected area is lost. This once again supports our observation that seed dispersal within the Avocado Belt is potentially more sensitive than pollen flow to the changes and permeability losses in the landscape matrix.

In summary, we found differences between patterns of connectivity related to seed and pollen dispersal, having a significant reduction in the degree of connectivity and switches

in the contribution of particular patches as the dispersal distance of pollen increased. In contrast, when the area weighed by the degree of conservation was considered, we only found a significant decrease of the IIC metric and the equivalent connectivity index. This highlights the importance of incorporating into connectivity analyses other attributes of patches, besides size, to have more realistic connectivity scenarios, as well as cost distances to incorporate the configuration of landscapes. Connectivity metrics tend to perform better when patch quality is considered, especially for landscapes with a heterogeneous habitat quality and spatially aggregated patches of good quality [84], which was the case with the Avocado Belt. We included as a proxy of habitat quality the degree of conservation of patches, but future studies may consider specific microclimatic and soil conditions that are relevant for the establishment and growth of tree seedlings and other plants [85,86]. The dispersal scenarios that we modeled used some of the most probable maximum pollen and seed dispersal distances, allowing us to encompass the intrinsic variation in dispersal capacities of the tree species that compose these temperate forests. However, for future studies, it is important to consider the limitations of our connectivity models and provide a fair interpretation. For example, many factors associated with landscape heterogeneity may reduce or increase dispersal distances, such as wind direction, humidity, and the abundance of dispersers.

Finally, the land-cover and land-use map that we developed had a good overall accuracy (0.69). However, this accuracy can be improved, especially for agriculture and settlement classes, using other classification frameworks. For example, increasing the number of training data for these classes and accounting for their spatial autocorrelation, since those are sources of errors that affect the classification accuracy [87].

*Implication for Conservation*

Decision-making to maintain landscape connectivity for native forests must consider the selection sites for conservation and restoration according to two criteria: the best individual sites that have attributes (e.g., habitat area, quality) that provide good resources for the establishment and reproduction of organisms, and sites that enhance the ecological connectivity and spatial cohesion of landscape networks [66]. We found that patches that support the connectivity of seeds and pollen have different characteristics, thus these attributes may be considered for proposing conservation and management strategies. In Michoacán, medium (>500–10,000 ha) to large size (>10,000 ha) patches located around the area with the highest density of avocado orchards provide habitat, and the small (<500 ha) patches act as stepping-stones for supporting the possible dispersion of seeds. At the same time, large and well-connected patches, distributed across the Avocado Belt, supported the interpatch connectivity and dispersion of pollen among patches. Therefore, both characteristics of the patches should be chosen if maintaining forest connectivity is to be maximized. However, the conservation of priority patches to maintain connectivity is more critical in the seed dispersal scenario since the amount of connected habitat ($ECA_f$) in that scenario is much smaller but would be crucial for the continued regeneration of forests. Furthermore, focusing on efforts to conserve patches that maintain seed connectivity is advantageous, as forest deforestation rates are faster than natural regeneration or assisted regeneration can act, and because forest decline due to climate change has been reported in the last 10 years on the Trans-Mexican Volcanic Belt [88]. Therefore, we propose that future research direction use prospective landscape modelling techniques and multi-temporal LULC maps to explore future landscape change trajectories and their effects on fragmentation and connectivity patterns of the temperate forest in the Avocado Belt, adding a potential decision-making tool.

The approach based on search ratios and iterations that we used to calculate the connectivity indices allowed us to simulate more realistic scenarios. Within the Avocado Belt, not all forest patches are connected, but rather they are small networks of connectivity that are interwoven depending on the dispersal capacity of seeds or pollen. In addition, with this approach, we were able to simulate a landscape for each patch and obtain

average values for landscape-level connectivity metrics. This approach can be applied in other landscapes when a clear patch delimitation is possible and the dispersal distance of individuals or gametes is well-established. Connectivity in the Avocado Belt is low, not only for the temperate forest remnants but also for other types of forests or animal species. Directing resource management and conservation activities is urgent to reduce deforestation and to begin to mitigate the negative impacts of fragmentation.

**Supplementary Materials:** The following supporting information can be downloaded at: https://www.mdpi.com/article/10.3390/land12030631/s1, Figure S1: Maps of the temperate forest within the Avocado Belt of Michoacán, Mexico, showing metrics of fragmentation; Figure S2: Maps of the temperate forest within the Avocado Belt of Michoacán, Mexico, showing the $IIC_f$ (a), $dIIC_f$ (b). Table S1: Classes of the land-use and land-cover map for the Avocado Belt in the state of Michoacán, Mexico; Table S2: Estimates of the mixed-effect models showing the effect of different dispersal capacities of trees (distance) and two attributes of patches (attribute) on connectivity patterns of the temperate forest in the Avocado Belt of Michoacán, México.

**Author Contributions:** Conceptualization: M.C.L.-C. and A.G.-R.; Methodology: M.C.L.-C., O.G.-G., A.G.-R. and E.Y.A.; Analysis and interpretation: M.C.L.-C. and A.G.-R.; Writing-original draft preparation: M.C.L.-C. and A.G.-R.; Writing-review and editing: M.C.L.-C., A.G.-R., O.G.-G., K.R.Y., A.D., E.Y.A., F.G.-O. and A.G. Funding Acquisition: A.D., A.G.-R., E.Y.A. and K.R.Y. All authors have read and agreed to the published version of the manuscript.

**Funding:** This research was funded by the ConTex (University of Texas System-CONACYT) collaborative program, grant 2020-28B "Avocado expansion in the forests of Michoacán: land use change and ecosystem services" to K.R.Y. and A.G.-R., and by PAPIIT-DGAPA-UNAM grant IN219223 "Genómica del paisaje de dos especies arbóreas en un sistema socio-ecológico dominado por el cultivo de aguacate" to A.G.-R.

**Data Availability Statement:** Not applicable.

**Acknowledgments:** We thank M. Sc. Diego Isla-López for technical support and the Laboratorio Nacional de Análisis y Síntesis Ecológica (LANASE, ENES Morelia, UNAM) for providing computing facilities; Andrés Piña-Garduño for spatial data assistance; M. Sc. Carmen Duque-Amado for helping in the first conceptual and programming development of the connectivity analysis. MCLC was supported by the ConTex grant 2020-28B (July 2021–April 2022) and by a postdoctoral scholarship from DGAPA-UNAM (staring in May 2022).

**Conflicts of Interest:** The authors declare no conflict of interest.

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
