# Peer review of "Estimating Fragmentation and Connectivity Patterns of the Temperate Forest in an Avocado-Dominated Landscape to Propose Conservation Strategies"

_land, doi:10.3390/land12030631_

Round 1
Reviewer 1 Report
I have read the interesting manuscript by Latorre-Cárdenas et al., in which they have presented the different techniques regarding the temperate fragments landscape. This is a well-written manuscript and the authors used different modern techniques for their investigation. I have following suggestions for further improvements.
1. In introduction authors can add the critical note on the economic benefits of Avocado and the species losses due to fragmentations.
2. A brief comparative about the techniques used.
3. In the discussion, the comparison can be added about the habitat fragmentation and its implications from it and also the impacts due to other studied crops in the similar regions.
Author Response
1. In introduction authors can add the critical note on the economic benefits of Avocado and the species losses due to fragmentations.
R/ We provided more critical information on the economic benefits of avocado cultivation and its effect on biodiversity loss. See lines 54 to 59. We also mentioned the effects of fragmentation and loss of connectivity on the decrease of species diversity due to fragmentation. See lines 73 to 78.
2. A brief comparative about the techniques used.
R/ We included in the introduction explanations about the graph theory, IIC metric and fragmentation metrics. See lines 82 to 91.
The application of different spatial analysis techniques was key to the development of this study and to obtain a comprehensive assessment of the conservation status of the temperate forests in the study area. On the one hand, the classification using random forests allowed us to obtain an accurate coverage map, which was a key input for subsequent spatial analyses. We used the classification to estimate fragmentation statistics, which allowed us to make an assessment of the landscape composition and to get an idea of the extent of habitat fragmentation and some of its more negative effects, such as the edge effect. Moreover, connectivity analysis using graph theory gave us an overview of the configuration of forest remnants. Taken together, each technique gave us information to make a robust spatial prioritization.
3. In the discussion, the comparison can be added about the habitat fragmentation and its implications from it and also the impacts due to other studied crops in the similar regions.
R/ We do not know studies, conducted in regions similar to ours, in which the impacts of other types of crops on the fragmentation and connectivity of the forest have been directly evaluated. Most of the studies carried out in crops such as coffee, palm oil, cocoa or soybeans mainly evaluate the effect on the diversity and composition of species that house the forest and not the same forest. We did not include these studies because they are not directly associated with the objectives of this study.
Reviewer 2 Report
Editor-in-Chief
Land
# land-2249477
Manuscript Title: Estimating fragmentation and connectivity patterns of the temperate forest in an avocado-dominated landscape to propose conservation strategies
The following queries have arisen through reviewing the manuscript. Moreover, I believe that the manuscript is well organized. There are some comments that can improve the strength of the paper. Therefore, a MINOR revision is recommended for the manuscript.
Details of the comments are as follows:
Abstract:
It is recommended to provide some numerical findings of the research in the abstract.
The main summary should be mentioned in the last part of the abstract.
Introduction:
Providing an explanation about different approaches regarding the available tools/methods for assessing structural and functional connectivity in the introduction can help to explain the method used.
In the introduction, provide some extra information on the landscape metrics-based methods in fragmentation analysis.
Provide explanations on evaluating changes in connectivity of landscape over time through spatial imagery techniques.
The research background section needs to be supported by more relevant sources.
Study area:
Mention some characteristics of the study area (temperature, precipitation, population density, dominant land use, proximity to population centers, roads and access pathways,…)
Research Methodology:
Indicate atmospheric and radiometric correction methods (if applied on satellite images).
Explain more about the feasibility of the developed scenarios.
Please mention some considered limiting factors in developing the scenarios.
Is there a size (area) threshold for forest patches in improving connectivity?
Discussion:
In the following sentence, indicate the range of area (approximately based on the research findings), The expression of big size and small size is a bit ambiguous:
“In Michoacán, medium-to-large size patches located around the area with the highest density of avocado orchards provide habitat, and the small patches act as stepping-stones for supporting possible dispersion of seeds.”
“At the same time, large and 491 well-connected patches, distributed across the Avocado Belt supported the interpatch 492 connectivity and dispersion of pollen among patches.”
In the last section, mention the sources of error and uncertainty in remote sensing data and methods.
Mention the limitations of the approach used (if your approach is used by other researchers in other areas).
The END
Author Response
Abstract:
It is recommended to provide some numerical findings of the research in the abstract.
The main summary should be mentioned in the last part of the abstract.
R/ We included in the abstract numerical findings and the main conclusions of the study. See lines 17 to 30.
Introduction:
Providing an explanation about different approaches regarding the available tools/methods for assessing structural and functional connectivity in the introduction can help to explain the method used.
R/ We provided in the introduction explanations about the graph theory, IIC metric and metrics based on it. Please see lines: 82 to 91.
In the introduction, provide some extra information on the landscape metrics-based methods in fragmentation analysis.
R/ We provided in the introduction a general explanation about metrics of fragmentation. Please see lines 89 to 91.
Provide explanations on evaluating changes in connectivity of landscape over time through spatial imagery techniques.
R/ We do not consider this to be necessary since assessing changes over time was not an objective of the study. Our aim to create the map was based on updating land use information. The most recent land use information for the region was for 2014, and this was used as one of the references for the creation of the training samples in the classification. This is mentioned in the lines 117 to 119, and 163 to 165.
The research background section needs to be supported by more relevant sources.
R/ We mentioned in the introduction recent studies that address the effects of the avocado cultivation on the biodiversity. See lines 54 to 59.
Study area:
Mention some characteristics of the study area (temperature, precipitation, population density, dominant land use, proximity to population centers, roads and access pathways,…)
R/ We provided the suggested characteristics of the study area. See lines 132 to 140.
Research Methodology:
Indicate atmospheric and radiometric correction methods (if applied on satellite images).
R/ We included in the methods section the atmospheric and radiometric correction methods that we used. See lines 155 to 159.
Explain more about the feasibility of the developed scenarios.
R/ We mentioned in the methods that seed and pollen dispersal scenarios are feasible because we built them based on genetic and restoration studies that assessed dispersal patterns of temperate forest tree species. See lines 213 to 214. However, we also mentioned in the discussion the limitations that these scenarios may have and the factors that can be considered in future studies. See lines 535 to 541.
Please mention some considered limiting factors in developing the scenarios.
R/ We mentioned in the discussions the limitations that these scenarios may have and the factors that can be considered in future studies. See lines 535 to 541.
Is there a size (area) threshold for forest patches in improving connectivity?
R/ It is difficult to have a single answer about the minimum size of patch area that improves connectivity. Forest patches contribute to landscape connectivity in different ways, depending on 1) their composition, 2) their configuration, and 3) the dispersal capacity and survival needs of the studied species. In the literature, we did not find an evaluation of the minimum habitat size needed to support a viable population of temperate forest’s species. As we mentioned (see lines 198 to 203) in the methodology, we discarded patches smaller than 10 ha for two reasons: 1) these patches have no core are, meaning that they could not provide available habitat and 2) to optimize the processing of spatial analyses, because including thousands of patches complicate the graphs construction and the interpretation of connectivity and fragmentation metrics.
Our model assumes that the larger the patch size, the greater the intra-patch connectivity and the greater the contribution to connectivity of the patch network as a receiving and sending source of flow (intra and flux fractions in our connectivity metric). We included this clarification in the lines 244 to 246.
Discussion:
In the following sentence, indicate the range of area (approximately based on the research findings), The expression of big size and small size is a bit ambiguous:
“In Michoacán, medium-to-large size patches located around the area with the highest density of avocado orchards provide habitat, and the small patches act as stepping-stones for supporting possible dispersion of seeds.”
“At the same time, large and 491 well-connected patches, distributed across the Avocado Belt
supported the interpatch 492 connectivity and dispersion of pollen among patches.”
R/ We provided the ranges area for small, medium and large patches in the highlighted sentences. See lines 555 to 558.
In the last section, mention the sources of error and uncertainty in remote sensing data and methods.
R/ We mention in the discussion the possible error and uncertainty of the classification. See lines 542 to 546.
Mention the limitations of the approach used (if your approach is used by other researchers in other areas).
R/ We mention along the discussion the limitation that the used approaches have (dispersal scenarios and supervised classification). See lines 535 to 541, 577 to 579, and 542 to 546.